# Physiological Analysis and Transcriptome Analysis of Asian Honey Bee (*Apis cerana cerana*) in Response to Sublethal Neonicotinoid Imidacloprid

**DOI:** 10.3390/insects11110753

**Published:** 2020-11-03

**Authors:** Jing Gao, San-Sheng Jin, Yan He, Jin-Hong Luo, Chun-Qin Xu, Yan-Yan Wu, Chun-Shen Hou, Qiang Wang, Qing-Yun Diao

**Affiliations:** 1Key Laboratory of Pollinating Insect Biology, Ministry of Agriculture, Institute of Apicultural Research, Chinese Academy of Agricultural Sciences, Beijing 100093, China; gaojing@caas.cn (J.G.); ssjinir@126.com (S.-S.J.); chunqinxu158@163.com (C.-Q.X.); wuyanyan@caas.cn (Y.-Y.W.); houchunsheng@caas.cn (C.-S.H.); wangqiang@caas.cn (Q.W.); 2National Maize Improvement Center of China, Beijing Key Laboratory of crop genetic Improvement, China Agricultural University, Beijing 100083, China; yh352@cau.edu.cn (Y.H.); jhluo@cau.edu.cn (J.-H.L.)

**Keywords:** *Apis cerana*, Imidacloprid, Detoxification, transcriptome analysis

## Abstract

**Simple Summary:**

In recent decades, there has been serious concern about the decline of honeybees in the world. One of the most debated factors contributing to bee population declines is exposure to pesticides, especially neonicotinoids. The most important Chinese indigenous species, *Apis cerana* presents a high risk on exposure to neonicotinoids, but few studies have explored the sublethal effects of neonicotinoids on *Apis cerana.* In this study, we highlight the molecular mechanism underlying the *A. cerana* toxicological characteristic against imidacloprid, the most commonly detected neonicotinoid in honey samples from *Apis cerana*. We not only investigated the physiological effects from sublethal doses of imidacloprid, but also identified several genes involved in a general stress response, including metabolism, catalytic activity, and structural molecule activity, response to stimulus, transporter activity, and signal transducer activity, as indicated by the GO analysis. In addition, genes related to the phenylalanine metabolism pathway, FoxO signaling pathway, and mTOR signaling pathway as indicated in the KEGG analysis were significantly up-related in the exposed bees. Overall, this study reveals the short-term sublethal effects of imidacloprid, which may be useful for accurately assessing the toxicity risk of Asian honeybees.

**Abstract:**

Asian honey bee (*Apis cerana*) is the most important Chinese indigenous species, while its toxicological characteristic against neonicotinoids is poorly known. Here, we combined physiological experiments with a genome-wide transcriptome analysis to understand the molecular basis of genetic variation that responds to sublethal imidacloprid at different exposure durations in *A. cerana.* We found that LC_5_ dose of imidacloprid had a negative impact on climbing ability and sucrose responsiveness in *A. cerana*. When bees were fed with LC_5_ dose of imidacloprid, the enzyme activities of P450 and CarE were decreased, while the GSTs activity was not influenced by the pesticide exposure. The dynamic transcriptomic profiles of *A. cerana* workers exposed to LC_5_ dose of imidacloprid for 1 h, 8 h, and 16 h were obtained by high-throughput RNA-sequencing. We performed the expression patterns of differentially expressed genes (DEGs) through trend analysis, and conducted the gene ontology analysis and KEGG pathway enrichment analysis with DEGs in up- and down-regulated pattern profiles. We observed that more genes involved in metabolism, catalytic activity, and structural molecule activity are down-regulated; while more up-regulated genes were enriched in terms associated with response to stimulus, transporter activity, and signal transducer activity. Additionally, genes related to the phenylalanine metabolism pathway, FoxO signaling pathway, and mTOR signaling pathway as indicated in the KEGG analysis were significantly up-related in the exposed bees. Our findings provide a comprehensive understanding of Asian honey bee in response to neonicotinoids sublethal toxicity, and could be used to further investigate the complex molecular mechanisms in Asian honey bee under pesticide stress.

## 1. Introduction

Pollinators, especial the honeybee species in the genus *Apis*, provide important services in agricultural ecosystems [1]. However, the honeybees have declined in population or contracted their distribution, posing a potential threat to the existence of species and global food security [2]. One of the most debated factors contributing to bee population declines is exposure to pesticides, especially neonicotinoids, which is considered as one of the possible stress factors that contribute to the Colony Collapse Disorder (CCD) [3].

Systemic neonicotinoids, such as imidacloprid, were widely used as a seed treatment to limit contact with non-target plants and insects. Neonicotinoids act as agonists of nicotinic acetylcholine receptors (nAChRs) by interfering with receptor signal transduction in the insect brain, leading to paralysis and mortality of target pests [4]. However, many studies have been indicated that widespread use of neonicotinoid insecticides is harmful to non-target insects, such as bees [5]. Due to the long-lasting ability and high photo stability of neonicotinoid insecticides, honeybees can be exposed to neonicotinoids through seed-dressed plants, foliar sprayed plants, and any plants that draw neonicotinoids from contaminated soil [6,7]. Both pollen and nectar in the hive were detected contamination by neonicotinoids [8]. However, honeybees cannot repel neonicotinoids in diets through their taste and olfactory system; instead, bees prefer to eat more food containing neonicotinoids (e.g., imidacloprid or thiamethoxam), even though the consumption of these neonicotinoid insecticides caused them to eat less food [9]. With the wide application of neonicotinoids in agriculture and horticulture, honeybees are increasingly at risk of exposure to these insecticides and thus contaminate the whole colony, as a consequence of foraging behavior in farmland or semi-forest habitat [10].

The traditional estimations of the harm by pesticides to honeybees normally relied on a measure of lethal dose (LD_50_), which only partially valuates the poisonous effects of pesticides. The reality is that even at very low doses, neonicotinoids have several lines of negative effects on honeybees, e.g., climbing ability, olfactory learning and memory, foraging activities, and reproduction [11,12,13]. In addition, exposure to sublethal dose of neonicotinoids can alter the honeybee immune system, thereby increasing pathogen loads and parasites infection [14,15,16]. Exposure of bees to one tenth of the LD_50_ dose of neonicotinoid clothianidin and imidacloprid can significantly enhance the proliferation of Deformed Wing Virus (DWV), one of the most common viruses infecting apiaries around the world [14]. Clothianidin acts as an additional stress factor by depressing the activation of the NF-κB (Nuclear factor kappa B) signaling pathway and exacerbates the negative effect of NF-κB transcription down-regulation in honeybees exposed to Varroa–DWV associated with honeybee immunity [14]. Furthermore, neonicotinoids significantly changed the energy metabolism of honeybees [17,18,19]. For instance, imidacloprid reduces lipid transport protein levels in honey bee hemolymph and induce oxidative stress in honeybees [17,20].

The sublethal effects of neonicotinoids have been focused on *Apis mellifera*, but little is known about native Asian honey bee species. Until 2015, only six studies demonstrated the effect of neonicotinoid pesticides on *A. cerana*, the most important Chinese indigenous species [5]. The risk of pesticides to *A. mellifera* cannot be completely inferred to other bee species [21,22,23,24]. A few comparative studies have shown that different neonicotinoids have different responses among *A. mellifera* and *A. cerana* [22,25,26]. Thus, it is difficult to draw a consistent conclusion about which species is more sensitive to neonicotinoid insecticides. Some studies tend to suggest that *A. mellifera* is the more sensitive to neonicotinoids than *A. cerana* when concerning the lethal toxicity [22,27]. However, this conclusion may not hold true for the sublethal effects of neonicotinoid exposure. A meta-analysis based on the oral acute LD_50_ and chronic LC_50_ in laboratory studies suggested *A. cerana* may have a higher sensitivity to pesticides (including neonicotinoids) than *A. mellifera* [21]. Different patterns of immune-related gene expression and enzymatic activities also indicated that *A. cerana* was more sensitive to sublethal doses of imidacloprid and clothianidin than *A. mellifera* [25].

As there is much evidence of the negative effects of neonicotinoids on honeybees, three highly toxic neonicotinoids (imidacloprid, clothianidin, and thiamethoxam) have been banned by the EU for outdoor use since 2018 [28]. Nevertheless, these pesticides are still widely used in nectariferous plants in other countries, especially in Asia area. Imidacloprid was the most commonly detected neonicotinoid in fruits and vegetables with 66 % detection in dominant apicultural provinces in China [29]. In addition, the latest research shows that imidacloprid was also the most frequently detected neonicotinoid in honey samples from *A. cerana* [30]. The information reminds us that *A. cerana* presented a high risk on exposure to imidacloprid, but few studies have explored the sublethal effects of imidacloprid on *A. cerana.* In this study, we determined the climbing behaviors, sucrose responsiveness, and detoxifying enzyme activity of *A. cerana* exposed to LC_5_ dose of imidacloprid. Then the *A. cerana* exposed to imidacloprid at 1, 8, and 16 h were subjected to RNA sequencing (RNA-Seq) aiming to identify the transcriptomic changes. This work will contribute to a better understanding of the molecular basis of major changes happening during imidacloprid exposure.

## 2. Materials and Methods

### 2.1. Insects Material

All colonies of *A. cerana* used in this study were obtained from an apiary on the campus of Yunnan Agricultural University, Kunming (E 103°40′, N 24°23′) from May to October 2015. Brood frames were placed in an incubator (33 ± 1 °C, 60 ± 10% relative humidity, darkness) after the cells were capped at approximately 9 days. Newly emerged workers were removed randomly in rearing cages (15 × 15 × 10 cm) within 12 h of their emergence. A total of 40 bees were used per cage. Bee rearing protocol used here was according to previous study [31] with minor changes. In short, 36 cages were placed in an incubator in darkness at 33 ± 1 °C and 50 ± 10% relative humidity. Two plastic feeders with eight separate feed tubes (300 μL each) were inserted into the cages vertically and changed daily [31]. Four tubes in one feeder are filled with deionized water, and eight tubes in the other feeder are filled with sucrose solution (50% wt/wt) and pollen (collected from the apiary placed in the forest and no chemical pesticides were applied at the pollen collection time). Bees were feed with food in the first week and then subjected to pesticide exposure as described below.

### 2.2. Pesticide Exposure Experiment

Imidacloprid (purities of all > 94%) was purchased from Dr. Ehrenstorfer, German. Stock solutions were prepared by dissolving the powder in acetone (purity > 99.99%) and then diluted with sucrose solution (50% wt./vol.). The final concentration of acetone in the sucrose solutions was equal to 0.03% (vol./vol.). The acute toxicity of imidacloprid to 7-day-old honey bees was determined in a 24-h feeding bio-assay. Eighteen cages (three replicates for each concentration including control, a total of 720 bees) were randomly selected to measure the survival rate. An additional 12 cages were used to measure the climbing ability and sucrose responsiveness, whereas the remaining cages were used for RNA-seq detection.

Before feeding assay, tested bees were starved for 4 h, and then fed with a 50% sucrose solution containing five concentrations (1.8 mg/L, 2.5 mg/L, 3.5 mg/L, 5 mg/L, 7 mg/L) of imidacloprid, and a solvent control. Mortality was recorded after bees were exposed via feeding to imidacloprid-containing sugar solution for 24 h. By calculating the median lethal concentration (LC_50_) of imidacloprid with Polo Plus 2.0 (Leora software, Memlo Park, CA, USA), the LC_50_ value of imidacloprid was 3.044 mg/L. To mimic the acute sublethal effects of the imidacloprid from contaminated beebread, we applied LC_5_ (0.968mg/L) of imidacloprid to 50% sucrose as the treatment for the following experiments, which is close to the maximum level (0.912 mg/L) in pollen obtained from bee hives [32].

For the enzyme activity analysis, bees were fed with LC_5_ of pesticide and collected at 1 h, 2 h, 4 h, 8 h, 12 h, 16 h, and 20 h. Control honeybees were fed with 0.03% acetone-containing sucrose syrup. For each time point, three replicates and three bees for each replicate were employed. An additional 12 cages were used to measure the climbing and sucrose responsiveness assay, whereas the remaining bees were used for RNA-seq assay. 

### 2.3. Climbing Assays

The climbing assays setup used here has been previously described in [31]. After exposure to imidacloprid-containing sugar solution for 24 h, ten bees from each cage were moved into an empty plastic box (10 cm × 5 cm × 15 cm) (except for the glass front allowing observation) with a mark placed 10 cm from the bottom and a fluorescent lamp in the top. The tests were performed in dark and started with the lamp turned on. The light stimulated locomotion of the bees by positive phototaxis. Before starting timer, the bees were gently knocked to the bottom of the box by tapping on the top, and keep the box standing vertically. The number of bees that climbed above the 10 cm line was recorded after 15 s. Climbing ability was expressed as a percentage of the number of bees above the 10 cm line as compared to the total number of bees. The experiment was conducted five times.

### 2.4. Sucrose Responsiveness

Sucrose responsiveness was performed following the protocol described in Yang et al. [33]. Sugar solution was prepared with distilled MilliQ water and sucrose at concentrations from 0.1 to 30% (*w*/*v*). Honey bees for proboscis extension reflex (PER) tests were collected after 8 h, 16 h, and 24 h exposed to LC_5_ dose of imidacloprid. The bee was briefly placed on ice for 1 min for immobilization and then fixed in the plastic holder, and left for 30 s before the stimulation test started. Sucrose stimulation was performed with a soaked toothpick touching both of the bee’s antennae at the same time for 1 s, and the PER was recorded (1 if a bee extended her proboscis and 0 if she did not respond). Alternated water trials between each sugar solution were used to reduce the possible effect of sensory sensitization to antennal touch [34]. All stimulations were performed at 3 min intervals. To avoid invalid counts, bees that responded to water or did not respond to any test concentrations of sugar solution were discarded before the test. 

### 2.5. Enzyme Activity Assay

Abdomens collected form three bees were pooled as one sample and grinded with 40 mM sodium phosphate buffer (pH 7.0) to measure the activity of carboxylesterase (CaE) and glutathione S-transferase (GSTs). For cytochrome P450 monooxygenases (P450s) activity measurement, midguts from three bees were pooled as one sample and grinded with extraction buffer (0.1 M sodium phosphate buffer (pH 7.5), 1 mM EDTA (Ethylene Diamine Tetraacetic Acid), 0.1 Mm PMSF (Phenylmethanesulfonyl fluoride), 0.1 mM DTT (Dithiothreitol) and 10 % glycerol) at 4 °C. Three biological replicates were performed for each treatment. The crude extracts were centrifuged at 10,000 rpm for 10 min and the supernatant fractions were used to measure enzymes activity. All extraction procedures were conducted at 4 °C. Protein concentration was assayed using bovine serum albumin as the standard.

GST activity was assayed with 1-chloro-2,4-dinitro-benzene as the substrate as described by Tang and Chang [35]. Enzymes activities of GST were determined by mixing of 900 μL of 100 mM phosphate buffer (pH 6.5) containing 30 mM reduced L-glutathione (GSH, Sigma-Aldrich, St. Louis, MO, USA) and 30 mM of 1-chloro-2,4-dinitrobenzene (CDNB, Sigma-Aldrich, St. Louis, MO, USA) as a substrate. The optical density (OD) was continuously measured at 340 nm every 15 s for a total of 2 min on a Synergy HTX plate reader. One unit of activity corresponded to the quantity of enzyme conjugating 1 mmol of GSH per min.

CaE activity was assayed by spectrophotometer using the substrate α-naphthyl acetate (Sigma-Aldrich, St. Louis, MO, USA) according to the method of Chanda et al. [36] with slight changes. The assay medium (3.2 mL total) consisted of 1.8 mL of 30 mM α- naphthyl acetate (containing 0.3 mM physostigmine) and 1.4 mL of 40 mM phosphate buffer (pH 7.0) containing 1 mM EDTA. The reaction was stopped by adding the chromogenic agent (1% Fast Blue B Salt and 5% SDS mixed in the ratio of 2:5) after incubation at 30 °C for 15 min. The absorbance was measured at 600 nm. Specific activity was expressed as mmol of the α-naphthol/min/mg protein.

Activity of P450s was determined by method of Shimada et al. [37] using 7-ethoxycoumarin as a substrate. In brief, P450 activity was determined in reaction mixtures containing 100 mM Tris-HCl (pH 7.5), 10 mM 7-ethoxycoumarin, 10 mM NADPH and an appropriate volume of crude enzyme in a final volume of 1mL. Samples were incubated for 15 min at 37 °C while shaking at 220 rpm in an Environ Shaker (Lab-Line Instruments Inc. Dubuque, IA). The reaction was stopped by adding 10 μL of 15% trichloroacetic acid (TCA) and centrifuged at 10,000 rpm for 1 min. Finally, 450 μL of glycine-NaOH was added to the mixture and measure the fluorescence of 7-hydroxycoumarin at 456 nm. The specific activity was expressed as pmol of 7-hydroxycoumarin formed/min/mg protein.

For all enzymes, blanks (reaction mixture free of crude enzyme sample) were periodically checked for non-enzymatic activities. Relative activities of each enzyme were calculated and expressed as ratio (fold ± SE) of the activity of the pesticide treatment to the activity of control before the pesticide exposure. All tests were performed in triplicate.

### 2.6. RNA Isolation and RNA Sequencing Analysis

Honey bees that were exposed to LC_5_ imidacloprid (for 1 h, 8 h, and 16 h) and the control group were randomly selected from three cages at each time point for transcriptomic analysis. Five bees from the same cage were pooled together as one RNA replicate samples. A total of twelve RNA libraries were constructed, representing samples from imidacloprid-treated groups and the control group. The libraries were as follows: 1h_1, 1h_2, and 1h_3 are replicate libraries for the group treated with imidacloprid for 1 h; 8h_1, 8h_2, and 8h_2 are replicate libraries for the group treated with imidacloprid for 8 h; 16h_1, 16h_2, and 16h_3 are replicate libraries for the group treated with imidacloprid for 16 h; and CK_1, CK_1, and CK_3 are replicate libraries for the control group as 1h comparison.

Samples were frozen and stored at −80 °C until the time of RNA isolation. Total RNA was extracted using TRIzol reagent following the manufacturer’s instruction. RNA quality was determined using the NanoDrop 2000 and RNA quantity was evaluated by Agilent 2100 RNA Nano 6000 Assay Kit (Agilent Technologies, Inc., Santa Clara, CA, USA). A total amount of 3 µg RNA per sample was used as input material for the RNA sample preparations. Sequencing libraries were generated using NEBNext^®^ Ultra™ RNA Library Prep Kit for Illumina^®^ (NEB, Ipswich, MA, USA) following manufacturer’s recommendations and index codes were added to attribute sequences to each sample. Second strand cDNA synthesis was subsequently performed using DNA Polymerase I and RNase H. cDNAs were used for PCR amplication and sequenced with an Illumina Nextseq. 500 platform in a 150 bp paired-end pattern. 

### 2.7. Bioinformatics Analysis of RNA-seq Data

Raw reads were pre-processed to remove low quality regions and adapter sequences. Sequence tag preprocessing was performed according to a previously described protocol with some modification. Reads with adaptors of low quality (>50%) or a high proportion of unknown bases (>5%) were removed. Clean data were mapped to the *A. cerana* genome (https://www.ncbi.nlm.nih.gov/genome/?term=Apis+cerana) using TopHat software with a maximum allowance of 2 nucleotide mismatches. 

The gene expression level was normalized using the fragments per kilobase of transcript per million mapped reads (FPKM) method [38]. Differential expression analysis of two groups was performed using the DESeq R package (1.18.0). The resulting *p*-values were adjusted using the Benjamini and Hochberg’s approach for controlling the false discovery rate. Genes with corrected *p*-values < 0.05 and|log2 (Fold change)| ≥ 1 were set as the threshold for significantly differential expression. Further, functional classification of the DEGs within up-regulated or down-regulated clusters was carried out using WEGO software [39], and KEGG pathway annotation by Blastall software against the KEGG (Kyoto Encyclopedia of Genes and Genomes) database (http://www.kegg.jp/). To determine the separation of expression patterns across samples, principle component analysis (PCA) and heatmap and cluster analysis (HCA) were carried out using the OmicShare tools, a free online platform for data analysis (www.omicshare.com/tools). Gene expression data for trend analysis (from 1 h to 16 h of *A. cerana* treatments) were normalized to log2(CK/CK) log2 (1h/CK), log2 (8h/CK), log2 (16h/CK). The trend profiles were generated by the Short Time-series Expression Miner (STEM, version 1.2.2b) software [40].

### 2.8. Confirmation of RNA-Seq Results by qRT- PCR

Total RNA was extracted from pooled honey bee samples, and first-strand cDNA was synthesized from 2 mg of total RNA using the Superscript II Kit (Invitrogen, Carlsbad, CA, USA) with oligo d(T)18 primers according to the manufacturer’s instructions. Quantitative RT-PCR reaction contained 100 ng cDNA, 1 pmol of each primer, 2x Sybr Green PCR buffer (Bio-Rad, Hercules, CA, USA). The PCR conditions for the amplification of and actin were as follows: 1 min at 94 °C, followed by 30 cycles of 45 s at 94 °C, 60 s at 54 °C, and 75 s at 72 °C. The PCR products were examined according the 2-∆∆CT method. The relative expression was calculated against that of the house keeping gene β-actin (GenBank accession number: HM640276). Each sample was assayed three times. Primer pairs used for each gene are listed in Appendix A.

### 2.9. Statistical Analyses

One-way analysis of variance (ANOVA) was used for comparing responses between treatment groups for climbing ability, syrup consumption, and gene expression (qRT-PCR). Statistical significance was analyzed using the Benjamini & Hochberg method comparision post-ANOVA test. IBM SPSS Statistics 19 (IBM Corp., Armonk, NY, USA) was used to conduct all statistical analyses. *p* < 0.05 was considered statistically significant.

## 3. Results

### 3.1. Climbing Ability, and Sucrose Responsiveness Changes of A. cerana after Exposed to Imidacloprid

Previous studies have shown that pesticide contaminants can influence bees’ behavior, such as climbing ability [41]. In this study, we firstly investigated the effects of imidacloprid on the geotactic climbing ability of *A. cerana*. After received the sublethal dose of imidacloprid for 24 h, very few bees (1–2 bees per cage at maximum) died after the climbing test. The percentage of bees above the line that had been treated with LC_5_ imidacloprid was nearly 100% at 1 d. The climbing ability was significantly impaired by LC_5_ imidacloprid at 3 d and 4 d (Figure 1A).

We next examined the effects of imidacloprid on sucrose responsiveness (Figure 1B). Pairwise comparisons showed that 24 h exposure had a significant effect on PER rate from 3% sucrose and higher. Only at 30% sucrose, bees exposed to pesticide for 16 h responded significantly less than those in CK group (*p* < 0.05). No obvious difference was observed among 8 h imidacloprid treatments compared to the control groups.

### 3.2. Influences of Imidacloprid on Detoxification Enzyme Activities

The enzyme activities of three major detoxification enzymes (GSTs, CaE, and P450s) from abdomen or midgut tissue of *A. cerana* were analyzed at eight indicated time points after exposure. During the experiment, no honey bee mortality was observed. After exposure of honey bees to LC_5_ of imidacloprid, P450 and CaE achieved the highest activity at 4 h (1.46 and 1.27-fold more than control, *p* < 0.05), and then the enzyme activities were sharply decreased after exposure with the time duration (Figure 2A,B). Unlike CaE and P450 activities, the GSTs activity was not influenced by the pesticide exposure (Figure 2C). This result suggested that GSTs play less roles in the detoxification.

### 3.3. Illumina Sequencing and Transcriptome Assembly

We analyzed the time-dependent changes in transcripts during toxicity effects of imidacloprid by using a high-throughput RNA-Seq analysis. Honey bees were exposed to LC_5_ dose of imidacloprid at three time points (1 h, 8 h and 16 h). Three biological replicates were performed by using RNA-seq sequencing for honey bee samples from pesticide treatment and control groups. The major sequencing assembly information is summarized in (Appendix A). About 7.12–8.44 Gb sequencing data of raw reads were obtained from twelve cDNA libraries. After filtering the raw reads, approximately 51.53 million clean reads (ranging from 47.7 to 54.95 million) were generated with an average GC content of 38.18%. The Q30 of the 12 samples were 86.88~89.43% with an average of 88.27%. Of these total clean reads, at least 87.32% were successfully matched either to a single or multiple genomic location for each library. In total, 12,138 genes were identified by sequencing analysis. A Pearson correlation analysis showed that the R^2^ coefficients between each group of two biological replicates were greater than 0.95, (Appendix A). Further, PCA analysis showed that imidacloprid treated groups and CK group clustered into four separated groups, moreover, 16 h group differed most from that in CK (Appendix A). Taken together, these results demonstrate the stability and reliability of the RNA-Seq results.

### 3.4. DEGs in Response to Sublethal Concentration of Imidacloprid

The level of annotated *A. cerana* genes was calculated on the basis of fragments per kilo base of exon per million fragments mapped (FPKM) metric (Appendix A). Differentially expressed genes (DEGs) were considered to be significant if Padj < 0.05 and|log2 (Fold change)| ≥ 1. Using CK group as a comparison, a total of 709 differentially expressed genes (DEGs) were obtained in different imidacloprid-treated groups. Of the differentially expressed genes, 430 unigenes received informative annotations, while 279 unigenes annotated as “uncharacterized protein”. The numbers of both up- and down-regulated DEGs were increased as exposure time going (Figure 3). Compared to CK, there are 16 up-regulated genes and 27 down-regulated genes in 1 h imidacloprid exposure treatment bees. In the 8 h vs. CK group, 189 DEGs were up-regulated and 126 DEGs were down-regulated. In the 16 h vs. CK group, 342 and 274 genes were up-regulated and down-regulated. According to the Venn analysis, we found 1, 49, and 201 genes were uniquely up-regulated, while 8, 30, and 184 genes were exclusively down-regulated expressed in each comparison, respectively. Moreover, 10 and 11 DEGs were commonly up-regulated or down-regulated among all comparisons, including cytoplasmic polyadenylation element-binding protein 1, vitellogenin, and HSP 83 (Appendix A). Compared with the 1 h group, 115 and 346 DEGs were identified in the 8 h and 16 h groups, respectively. In both comparisons, the number of up-regulated genes was more than those of down-regulated genes (Appendix A). Compared with the 8 h group, 16 DEGs were upregulated and 17 were downregulated in 16 h group (Appendix A).

### 3.5. Trend Analysis of DEGs

To further investigate the gene expression pattern of all DEGs, a trend analysis of the DEGs was performed using the STEM software. As shown in Figure 4, 709 DEGs that were obtained by comparison with imidacloprid-treated groups and the CK group were categorized in 17 profiles. The profile information of the DEGs by this focused trend analysis are shown and annotated in Appendix A. DEGs were significantly enriched in six profiles with a Q value of ≤ 0.05, including three up-regulated patterns (profile 10, 12, and 17) and three down-regulated patterns (profile 1, 7, and 9) (Figure 4).

Profile 17 (171 DEGs) and profile 1 (141 DEGs) had the largest numbers of DEGs whose abundance continuously up-regulated or down-regulated with the imidacloprid exposure, indicating that the changes in expression of individual genes occurred at an early exposure stage in these profiles. Although a large number of genes are clustered in profile 17, over half of the genes (95 DEGs) are not clearly annotated. Fourteen DEGs annotated as transporter and receptor proteins, such as organic cation/carnitine transporter, odorant receptor, and voltage-dependent calcium channel, were found in profile 17. In profile 1, 34 unigenes annotated as “uncharacterized protein”. Among the well annotated genes, 23 unigenes participating in metabolic processes were found in this profile, such as genes encoding UDP-glucuronosyltransferase, NADH dehydrogenase, and 3-hydroxyacyl-CoA dehydrogenase. In addition, genes encoding structural proteins, i.e., cuticle protein, and endocuticle structural glycoprotein were also included.

The genes in profile 7 (59 DEGs) and profile 12 (94 DEGs) had the opposite expression pattern. The expression changes of these genes occurred at intermediate stage of exposure in this experiment. As a down-regulated pattern profile, in addition to the genes encoding metabolic enzymes (13 DEG) and structural proteins (4 DEG), four genes related to neural development were also found in profile 7. Genes relating to the cellular immune pathway including those for AMPs (antimicrobial peptides), esterases, serine protease, lachesin, and leukocyte elastase inhibitor were clustered in profile 12, which exhibited a significant increase after 8 h exposure and then keep the similar expression level after 16 h treatment.

The genes in profile 9 and profile 10 were responsive to imidacloprid toxicity at a late stage of exposure in this experiment. The genes included in profile 9 (41 DEGs) showed a relatively lower expressed level at 16 h exposure stage. In this profile, the greatest changed genes were observed for encoding adenosine deaminase CECR1-like, ADP-ribosylation factor GTPase-activating protein, and collagen alpha-1(I) chain. By contrast, in profile 10 the expression levels of DEGs were relatively higher at 16 h than in the other two time point. Genes showing the greatest changes in expression in this profile included those for transfer RNA alanine, serine protease inhibitor, and odorant receptors.

### 3.6. Functional Annotation and Classification of Differentially Expressed Genes

The 709 DEGs were categorized into three main categories including biological process, cellular component and molecular function (Appendix A). A total of 49 functional groups were generated according to sequence homology. According to GO classification, biological process accounted for the majority of GO categories (40.8%), followed by cellular component (34.7%) and molecular functions (24.5%). Among these groups, “metabolic process”(GO:0008152, 208 genes, 29.3%), “cellular process”(GO:0009987, 193 genes, 27.2%), “single-organism process”(GO:0044699, 177 genes, 25.0%), “membrane”(GO:0016020, 140 genes, 19.7%), “binding” (GO:0005488, 216 genes, 30.5%), and “catalytic activity” (GO:0003824, 173 genes, 24.4%) were the most abundant terms. Possibly because of the small number of unigenes, only one GO category (odorant binding, GO:0005549) was significantly enriched after correction for an FDR (<0.05).

To further evaluate the effectiveness of the annotation process, the DEGs within up-regulated patterns (310 unigenes in profile 10, 12, and 17) or down-regulated patterns (241 unigenes in profile 1, 7, and 9) (with Q value < 0.05) were also subjected to GO term analysis. In some GO terms the number of down-regulated DEGs was more than the number of up-regulated DEGs, such as “metabolic process” and “cellular component organization or biogenesis” in the biological process category, and “binding”, “catalytic activity”, and “structural molecule activity” in the molecular functions category. By contrast, more up-regulated DEGs were enriched in “response to stimulus”, “localization” and “multicellular organismal process” in biological process category, and also in the class “transporter activity”, “molecular transducer activity”, and “signal transducer activity” in the molecular functions category (Figure 5 & Appendix A).

KEGG pathway enrichment analysis was also conducted with the DEGs from aforementioned cluster groups. The top 20 KEGG pathways with the highest representation of DEGs within the up-/down-regulated clusters are shown in Figure 6 & Appendix A. For DEGs within the down-regulated cluster groups, the enriched pathway was mostly associated with protein process and metabolism, such as “Glycine, serine, and threonine metabolism”, “Arginine and proline metabolism”, “Protein processing in endoplasmic reticulum”, albeit not significantly (*p* > 0.05). In the up-regulated cluster groups, the significantly enriched pathways involved protein processing in “Phenylalanine metabolism”, “Tyrosine metabolism”, “FoxO signaling pathway”, and “mTOR signaling pathway”.

Additionally, immune-related pathways were selected for further analysis based on the result of KEGG pathway analysis. A total of 33 DEGs were assigned to the cellular immune-related pathways, including APMs, lysosome, ribosomal biosynthesis, Toll/TLR signaling pathways and Jak/STAT signaling pathway (Table 1). In the ribosomal biosynthesis pathway, five out of eight DEGs displayed down-regulated trends. By contrast, the expression levels of the six AMPs, lachesin, and leukocyte elastase inhibitor were significantly increased at 8 h after exposure. Most genes in the Toll/TLR pathway were classified in up-regulated trend profiles (e.g., AMPs, serine protease, protein toll), demonstrating that most genes involved in Toll/TLR signaling pathways were activated in response to imidacloprid toxicity.

According to the behavior test described above, acute exposure to LC_5_ imidacloprid depressed the sensitivity to sucrose in *A. cerana* (Figure 1). It is apparent that sublethal dose of neonicotinoids induced olfactory dysfunction of bees. Therefore, we also focus on the expression changes of genes functions in insect chemosensation. Functional annotation of DEGs revealed that several chemosensory related genes showed different expression level pattern at exposure duration, including three general odorant-binding protein (OBP) genes, a gustatory receptor (GR), ten odorant receptor (OR) genes, and two members of the ancient ionotropic glutamate receptor family (Table 2). The expression levels of *ASP1*, *OBP13*, and *OBP59a* exhibited down-regulated pattern during the imidacloprid exposure; in addition, the expression levels of most odorant receptors (7 out of 10) in imidacloprid-treated bees were significantly higher than that in control bees at 16 h after exposure. 

### 3.7. Validation of RNA-seq Data by Quantitative Real-Time PCR

To confirm the results observed during RNA-seq analysis, we performed qRT-PCR analysis on 16 selected DEGs, including gene encoding cytochrome P450, abaecin, NF-kappa-B inhibitor cactus-like protein, and calcium release-activated calcium channel protein 1, etc. Total RNA was isolated from the treated and the un-treated samples (CK) using the same exposure protocol as RNA-seq assays. Most qRT-PCR expression patterns showed similar expression patterns to those obtained by RNA-Seq assay, though the fold change of expression patterns from RNA-seq and qRT-PCR diverged. Only two DEGs (major royal jelly protein 5-like and NF-kappa-B inhibitor) were inconsistent. Thus, the qRT-PCR results confirmed that the RNA-Seq data were reliable (Figure 7).

## 4. Discussion

Previous studies have shown that neonicotinoids have negative impacts on various bee species, but only a few studies investigated *A. cerana*, the most important native honeybee in China [5]. Due to the wide use of imidacloprid in China, it is indispensable to estimate the effects of imidacloprid on *A. cerana*. We show for the first time that dynamic genetic variation of *A. cerana* responds to acute toxicity of imidacloprid. The comparison between control and exposed worker *A. cerana* emphasizes that a sublethal dose of imidacloprid led to a growing imbalance of transcriptome as the exposure time continued. In this work, the LC_50_ dose of imidacloprid is 3.044 mg/L, which is above the considered “field realistic” range as well as the estimated LC_50_ dose of *A. mellifera* (1.76 mg/L) according to a meta-analysis of fourteen published studies [42]. 

We found that short-term exposure to LC_5_ dose of imidacloprid reduced climbing ability and sucrose responsiveness in *A. cerana* (Figure 1), which is consistent with the previous studies on *A. mellifera* [43,44,45]. Along with the behavioral changes induced by imidacloprid exposure in *A. cerana*, changes in gene expression related to physiological responses to pesticides were also observed in the present study. We obtained a dynamic overview of the gene expression responding to a sublethal dose of imidacloprid in *A. cerana*. The transcriptional alternation induced by imidacloprid is reinforced by exposure time, and most differentially expressed genes (DEGs) were evident at 16 h after exposure (Figure 3). The enhancing effects of imidacloprid on transcriptome with time reflect the intensification of internal disorders and the insufficient resilience of exposed honeybees to chemical stress. In addition, the number of DEGs within up-patterns (310) was higher than that within down-patterns (241), suggesting more responsive genes were activated by imidacloprid. GO term enrichment analysis of these DEGs revealed that the most of enriched ontology terms was related to a general stress response. In exposed honeybees, more genes involved in metabolism, catalytic activity, cellular component organization or biogenesis, and structural molecule activity are down-regulated. By contrast, more up-regulated genes were enriched in terms associated with response to stimulus, transporter activity, and signal transducer activity. Additionally, genes related to the phenylalanine metabolism pathway, FoxO signaling pathway, and mTOR signaling pathway as indicated in the KEGG analysis were significantly up-related in the exposed bees (*p* < 0.05) (Figure 6 and Figure 7).

Metabolic processes are important regulators of physiological and behavioral performance in honey bees. In our study, the largest gene cluster affected by imidacloprid belonged to metabolic processes as indicated by the GO term enrichment analysis. Christen et al. have demonstrated that down regulation of genes associated with glycolysis and lipids metabolism are commonly observed in different neonicotinoid treatments [46]. Genes involved in glycolytic and sugar-metabolizing pathways are down-regulated upon imidacloprid exposure in adult bees, but the trend is conflicting with the larva [47]. Consistent with those results, imidacloprid exposure led to down-regulation of genes functioning in glycolysis (e.g., UDP-glucuronosyltransferase, glucose dehydrogenase) and lipids metabolism (e.g., lipase H-A-like, phospholipase) in *A. cerana* workers. Additionally, genes related to phenylalanine and tyrosine metabolism were increased by imidacloprid as indicated by the KEGG enrichment analysis (Figure 7), which may diminish insulin sensitivity [48,49]. The depressed glycolysis at transcriptional level may also be related to the overexpression of genes involved in insulin signaling pathway, such as insulin receptor substrate (LOC108004059, profile 17; LO107999616, profile 17), 3-phosphoinositide-dependent protein kinase (LOC107994871, profile 17) and insulin-like peptide receptor (LOC108000589, profile 12; LOC108002698, profile 15). Insulin signaling pathway can directly or indirectly interact with the mTOR signaling pathway and FOXO signaling pathways to regulate carbohydrate metabolism and energy restore. Moreover, the genes participated in energy metabolism (ATP synthase, NADH dehydrogenase, etc) were classified in the down-pattern profiles (profile 1, 7, 9). According to Nicodemo et al., imidacloprid prevents ATP synthesis, thereby inhibiting mitochondria respiration in honey bees [18]. Collectively, the less efficient carbohydrate catabolism and lower energetic metabolism is likely related to the negative impact of imidacloprid on behavioral performance that requires muscles working with high ATP consumption, e.g., flight [50]. Whether the impairment of the climbing ability observed in this study is directly induced by the lower energy supply needs further evidence for validation.

In honeybees, the activities of detoxification enzymes are appropriate biomarkers for evaluating the stress response of xenobiotics, such as pesticides [51]. Glutathione-S-transferase (GST), carboxylesterase (CaE), and cytochrome P450 (P450) are three main detoxification enzymes, which are chiefly responsible for the detoxification processes phase I (functionalization) or phase II (conjugation) in honeybees [25,52,53]. The bioassay time course experiment showed that LC_5_ concentration of imidacloprid depressed the activities of P450 after 4 h exposure (Figure 2). The decreased expression level of five P450 genes belonging to CYP3 subfamily (LOC108000928, profile 1; LOC 107999234, profile 1) and CYP6 subfamily (LOC108004038, profile 1; LOC108003993, profile 7; LOC108003970, profile 9) were observed in the transcriptomics data. The activity of CaE were also depressed after 4 h of exposure to imidacloprid (Figure 2). Although the expression of two venom carboxylesterases decreased after 16 h imidacloprid exposure (2.39 and −2.03 fold), statistical analysis show that the difference is not significant due to the FDR value greater than 0.064 and 0.073. Two venom carboxylesterases (LOC107994233 and LOC107998239) showed down-regulated (−2.39 and −2.03 fold) after 16 h imidaclopride exposure but not classified as down-regulated in the trend analysis due to the FDR value greater than 0.064 and 0.073. Whether the lower enzyme activity is involved in the depressed expression of P450s and CaE in *A. cerana* is questionable and requires further investigation. Wu et al. demonstrated that sublethal imidacloprid exposure induced the down-regulated expression of CYP450 family members in clades 6 and 9 during the larval stage [54]. Members in the P450 subfamilies CYP6 and CYP3, are most frequently involved in xenobiotic detoxification and evolution of the hormonal and chemosensory processes in bee species [55,56]. Manjon et al. have investigated the whole CYP3 clade of P450s from *A. mellifera* and demonstrated that CYP9Q1-3 exhibits capacity to metabolize imidacloprid with lower efficiency than that in thiacloprid [56]. Whether the differential expressed P450s identified in our RNA-seq analysis have detoxification functions or are simply involved in other metabolic pathways is questionable and requires further study. It is worthwhile to note that activation of the detoxification mechanism dominated by cytochrome P450 may require some time in honey bees. Previous studies showed that a cluster of genes encoding cytochrome P450 enzymes in clades CYP4, CYP6, and CYP9 were significantly up-regulated after exposure to imidacloprid at later time points [47,57]. The down-regulation of detoxification enzymes after short-term imidacloprid exposure can make bees more sensitive to neonicotinoids, consequently providing a protective effect for subsequent exposure [58]. A comparison between *A. mellifera* and *A. cerana* revealed that Glutathione-S-transferase (GST) activity was significantly elevated in *A. mellifera* at 48 h after exposure to imidacloprid, but no significant change was observed in A. *cerana* [25]. Consistent with this result, no obvious alternation in the activity or transcriptional change of GST was observed in imidacloprid-exposed bees, suggesting that GSTs might not be the dominator in imidacloprid detoxification in *A. cerana*.

Adverse effects of neonicotinoids on bee immune systems were previously reported [14,59]. Numerous studies have suggested that sublethal doses of imidacloprid caused transcriptional alterations involved in the immune system in *A. mellifera* and bumblebee *Bombus* [59,60,61]. The immune system of *A. cerana* comprises more genes than *A. mellifera*, indicating a stronger protecting ability against xenobiotics in *A. cerana* [62]. The Toll/TLR pathway is important for innate immunity in insects by regulating the immune-responsive genes. As key components of the Toll pathway, the antimicrobial peptides (AMPs) response to chemical stress or bacterium via the rapid expression and effective delivery to the site of infection in honeybees [63]. According to our data, antimicrobial peptides including defensin1 (LOC107993803), defensin2 (LOC108000415), apidaecin (LOC108000468), abaecin (LOC108002218), and hymenoptaecin (LOC107993492), hymenoptaecin pseudogene (LOC108001747) were more steeply increased at 8 h post treatment. The similar results were also reported by Li et al. [25], which demonstrated that defensin2, abaecin, and hymenoptaecin were significantly upregulated in *A. cerana* exposed to imidacloprid after 24, and 48 h. Distinct expression patterns of antimicrobial peptides were observed between *A. cerana* and *A. mellifera* throughout the neonicotinoids exposure period, exhibiting their different abilities relating to innate immune response against neonicotinoids [25]. Though esterases appear to be less important in detoxification of honey bees, they usually serve as AMPs and induce an immune response in pesticide-exposed insects [64]. The expression pattern of three esterases (LOC107998416, LOC107998417, LOC107998473, profile 12) in exposed bees was very similar to AMPs in the current study. It is known that neonicotinoid clothianidin exposure interferes with NF-κB activation and subsequently affects the induction of AMPs [14]. In this study, two genes encoding the NF-kappa-B inhibitor (LOC107996104 profile 16, LOC108002391 profile 16) were found to be up-regulated in exposed bees at 8 h post imidacloprid exposure and then their expression slightly decreased. The negative impact on NF-κB activation could affect the stress responses of the honeybees to the pesticide via regulating immune-related genes, which acts as the downstream effectors activated by Toll pathways [65]. The increasing expression of the genes in the immune system with exposure time would reflect the ability of bees to recover from short-term pesticide toxicity. However, reduction in the expression of AMPs after long-term exposure to imidacloprid was reported in *A. mellifera* [54,66]. This temporal variance implied that rapid response of the immune system can make *A. cerana* more resistant to imidacloprid at initially, nevertheless, the immunosuppression was raised after prolonged exposure to prevent damages deriving from excessive immune reaction.

It has been demonstrated that sublethal doses of neonicotinoids impaired olfactory associative function such as foraging behaviors, olfactory learning, memory of honeybee [12,43,67]. Insects recognize odor molecules in the external environment via a series of proteins implicated in olfactory recognition system, including odorant-binding proteins, chemosensory proteins, and chemoreceptors. Insect odorant-binding proteins (OBPs) binding to odorant molecules and transport them through the sensillar lymph to the membrane to activate olfactory receptors [68]. Li et al. [69] have demonstrated that binding between imidacloprid and odorant-binding protein ASP2 in *A. cerana* induced a conformational change that decreased the binding affinity of ASP2 to floral vinolatile β-ionone. The expressional changes of three Obp genes identified in this study in line with existing reports in which Obp genes were depressed by a sublethal dose of imidacloprid and other neonicotinoids in *A. mellifera* [46,54,66,70]. The downregulation of Obp genes implied a decreased chemosensory ability in exposed bees. Interestingly, we found that members in the classes of insect chemosensory receptors (odorant receptor, gustatory receptor, ionotropic glutamate receptor) showed up-regulated trends from Obps (Table 2). Similarly, up-regulation of chemosensory receptors induced by sublethal doses of imidacloprid were also observed in *Aphidius gifuensis* [71]. Odorant receptors (ORs) belong to chemoreceptor superfamily and act as the primary receptors for perception of odor chemical cues, but only a few studies described their functional characterization in honeybees. OR binding odorant molecules in honeybees may be task-dependent. In *A. cerana*, AcerOr1 is broadly sensitive to floral odorants compounds, while *AcerOr2*, which is orthologous to the co-receptor, did not responds to any floral odorants except its agonist VUAA1 [72]. AmOr11 specifically responded to the main queen retinue pheromone 9-oxo-decenoic acid (9-ODA) only when AmOr2 was coexpressed [73]. These results implied that the dysfunctional olfactory behaviors caused by imidacloprid require multiple molecular mechanisms underlying the olfactory recognition. 

## 5. Conclusions

*Apis cerana cerana* is the most important Chinese indigenous species that plays a critical role in maintaining the balance of regional ecologies and agricultural economic development. With the wide use of imidacloprid in China, it is indispensable to estimate the effects of imidacloprid on *A. cerana*. In this study, we found that exposure to a sublethal dose of imidacloprid significantly affected the physiological performance of *A. cerana*, which is potentially caused by changes in gene expression involved in metabolic processes, detoxification and immune systems, and olfactory associative functions. Obviously, whether the transcription level of these genes is correlated to the protein level needs to be proved in the future. In addition, further studies are required to confirm the identified responsive genes involved in proximate mechanisms underlying the fast defense response. Our findings provide a comprehensive understanding of Asian honey bee in response to sublethal neonicotinoids toxicity, and could be used to further investigate the complex molecular mechanisms in Asian honey bee under pesticide stress.

## Figures and Tables

**Figure 1 insects-11-00753-f001:**
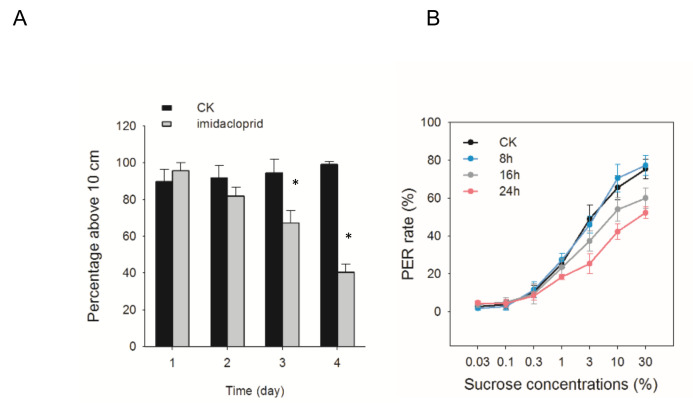
Sublethal effects of imidacloprid on climbing ability and sucrose responsiveness of *A. cerana*. (**A**) Geotaxis climbing ability of honey bees after imidacloprid exposure. The line above each bar is the SE. Significant differences to CK with *p* < 0.05 are indicated by an asterisk above the bars and were determined by two-way ANOVA followed by Bonferroni’s post hoc test (n = 10). (**B**) Sucrose responsiveness variation of *A. cerana* after Oimidacloprid exposure. From 3% sucrose and higher, PER rate (%) was significantly lower in bees exposed to imidacloprid for 24 than that of control group. Values are mean ± SEM (*N*_CK_ = 67, *N*_8h_ = 61, *N*_16h_ = 52, *N*_24h_ = 60 bees).

**Figure 2 insects-11-00753-f002:**
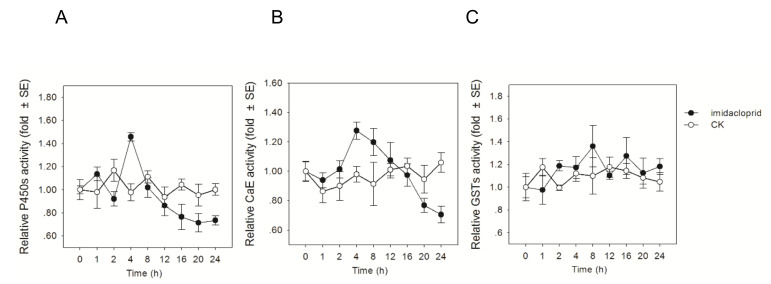
Variable activities of three detoxification enzymes of *A. cerana* workers after feeding treatments. (**A**–**C**), relative ratios of enzymes activity of P450s, CaE, and GSTs in bees after imidacloprid exposure. Data are means of three independent experiments, and error bars represent ± standard error (SE) (n = 3).

**Figure 3 insects-11-00753-f003:**
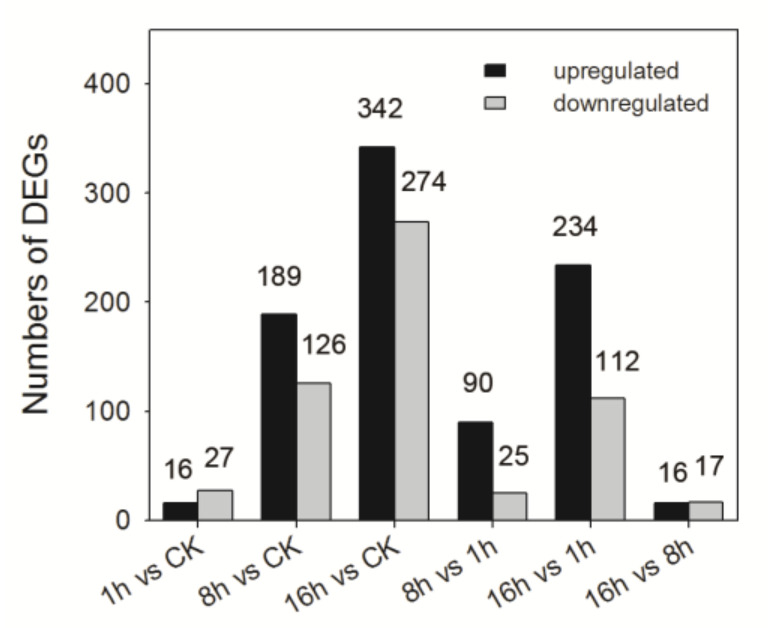
The Differentially expressed genes (DEGs) of *A. cerana* treated with imidacloprid at the indicated time points in the study (1 h, 8 h and 16 h). Up-regulated and down-regulated means that these genes were higher or lower expressed in treatment group compared to CK group, or these genes were higher or lower expressed in later time point compared to early time point.

**Figure 4 insects-11-00753-f004:**
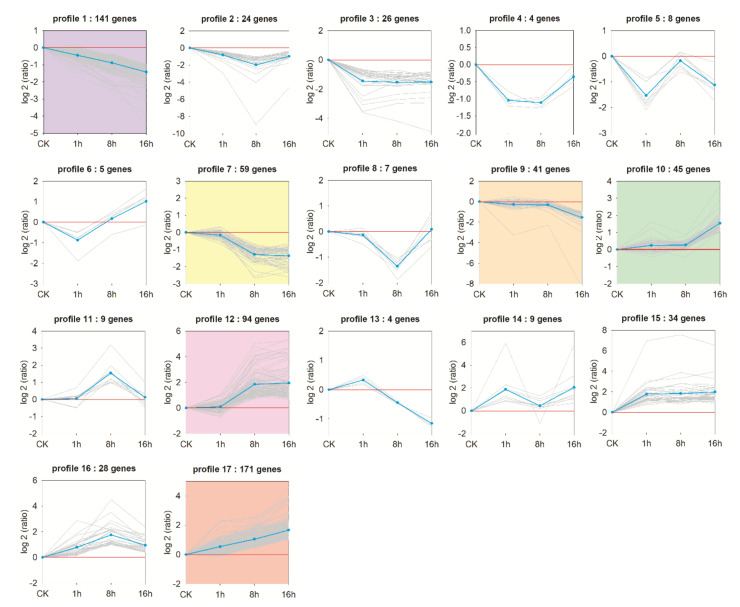
Trend analysis of the DEGs during imidacloprid treatments. The trend profiles are generated by STEM software. The blue line shows normalized level of gene expression in each profile and the grey line represents the individual gene belonged to this profile. The profile number and the number of genes belonging to each pattern is indicated above the profile. Profiles with a colored background indicate that gene expression was significantly enriched in this pattern, with a Q value of ≤0.05.

**Figure 5 insects-11-00753-f005:**
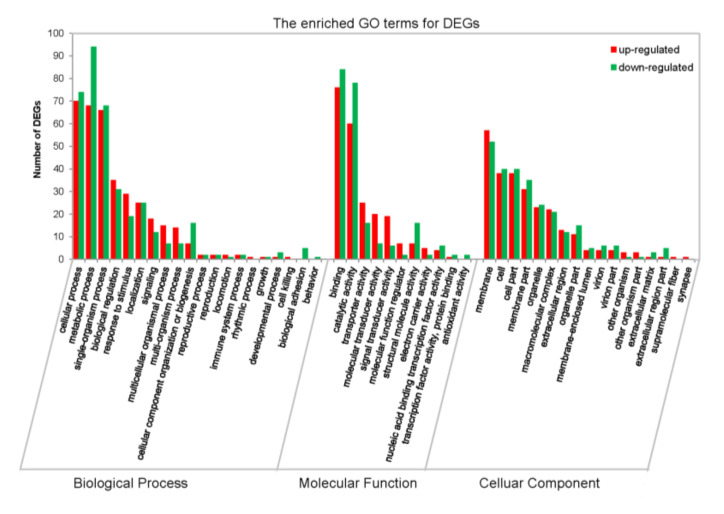
Gene ontology classification of DEGs with up- or down- regulated patterns. The up-regulated genes are union of DEGs in profiles 10, 12, and 17, and down-regulated genes are union of DEGs in profiles 1, 7, and 9. The red and green bars represented DEGs derived from up-regulated and down-regulated, respectively. The x-axis indicates the second term of gene ontology; the left y-axis indicates the number of genes involved in a GO term. See also Appendix A shown the whole GO enrichment analysis of 551 DEGs associated with the trend clusters.

**Figure 6 insects-11-00753-f006:**
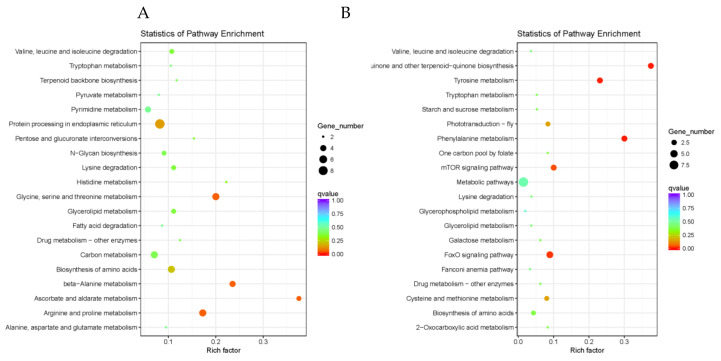
KEGG (Kyoto Encyclopedia of Genes and Genomes) pathway enrichment analysis of DEGs with up- or down- regulated patterns. (**A**) KEGG pathway enrichment analysis of DEGs with down-regulated profiles. (**B**) KEGG pathway enrichment analysis of DEGs with up-regulated profiles.

**Figure 7 insects-11-00753-f007:**
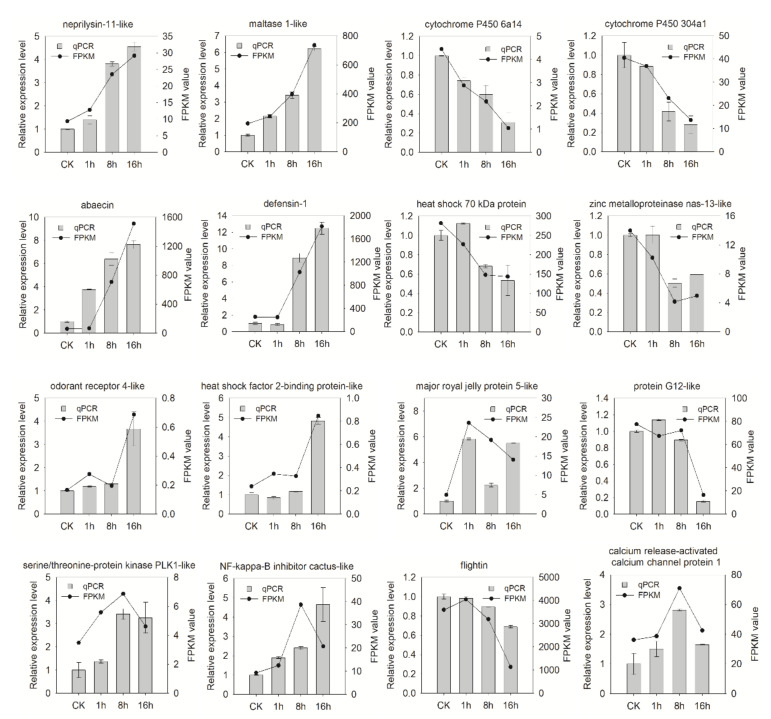
Verification of 16 differentially expressed genes by qRT-PCR. Both 16 different expressed genes in imidacloprid exposed bees were randomly selected for qRT-PCR validation. Columns and bars represent the means and standard errors of the qPCR results (Y axis at left). Scatters and lines represent the FPKM value of the transcriptome result (Y axis at right). The experiment was conducted three times with independent biological replicates. Note: neprilysin-11-like LOC107993231; maltase 1-like LOC107993621; probable cytochrome P450 6a14 LOC108004038; probable cytochrome P450 304a1 LOC108000928; abaecin LOC108002218; defensin-1 LOC107993803; heat shock 70 kDa protein cognate 4-like LOC107996313; zinc metalloproteinase nas-13-like LOC107994936; odorant receptor 4-like LOC107996745; heat shock factor 2-binding[protein-like LOC107996546; major royal jelly protein 5-like LOC107997174; NF-kappa-B inhibitor cactus-like LOC108002391; serine/threonine-protein kinase PLK1-like LOC107999607; flightin LOC107997838; protein G12-like LOC107999828; calcium release-activated calcium channel protein 1-like LOC107994288.

**Table 1 insects-11-00753-t001:** Expression profiles of the DEGs associate with cellular immune-related pathways.

Gene Name	Pathway	Gene Description	Profile
LOC108002218	AMP	abaecin	12
LOC108000468	AMP	apidaecins type 22	12
LOC107993803	AMP	defensin-1	12
LOC108000415	AMP	defensin-2	12
LOC107993492	AMP	hymenoptaecin	12
LOC108001747	AMP	hymenoptaecin pseudogene	12
LOC108004481	Jak/STAT	E3 ubiquitin-protein ligase TRIM71-like	17
LOC108000724	Jak/STAT	E3 ubiquitin-protein ligase MARCH2-like	1
LOC108003398	Ribosomal Biosynthesis	60S ribosomal protein L35	1
LOC107992528	Ribosomal Biosynthesis	60S ribosomal protein L37a	1
LOC108000385	Ribosomal Biosynthesis	60S ribosomal protein L39	2
LOC108001255	Ribosomal Biosynthesis	60S ribosomal protein L28-like	2
LOC107994696	Ribosomal Biosynthesis	ribosomal RNA processing protein 1 homolog	7
LOC108002849	Ribosomal Biosynthesis	60S ribosomal protein L37	8
LOC108003508	Ribosomal Biosynthesis	histone H2A-like	1
LOC107998816	Ribosomal Biosynthesis	histone H2A.Z-specific chaperone CHZ1-like	7
LOC107996866	Toll/TLR	toll-like receptor 4	7
LOC107998162	Toll/TLR	toll-like receptor 13	17
LOC108001172	Toll/TLR	protein toll-like	11
LOC107995659	Toll/TLR	protein toll	12
LOC107996104	Toll/TLR	NF-kappa-B inhibitor cactus-like	16
LOC108002391	Toll/TLR	NF-kappa-B inhibitor cactus-like	16
LOC107998380	Toll/TLR	serine protease inhibitor 3-like	10
LOC107994873	Toll/TLR	serine protease gd-like	12
LOC108002098	Toll/TLR	serine protease snake-like	12
LOC108004289	Toll/TLR	serine/threonine-protein kinase 32B	12
LOC107999607	Toll/TLR	serine/threonine-protein kinase PLK1-like	16
LOC108003108	Toll/TLR	serine/threonine-protein kinase Nek5-like	17
LOC107996922	IG Superfamily Genes	lachesin-like	12
LOC107995322	leukocyte	leukocyte elastase inhibitor-like	12
LOC107996108	Lysozyme	lysosomal alpha-mannosidase	9
LOC107999764	Scav. Receptor B	scavenger receptor class B member 1	1

**Table 2 insects-11-00753-t002:** Expression profiles of the DEGs associate with chemosensation.

Gene Name	Gene Description	Profile
LOC107996029	odorant-binding protein 59a-like	1
LOC108002840	general odorant-binding protein 69a-like (ASP1)	4
LOC107999620	general odorant-binding protein 56d-like (OBP13)	9
LOC107996745	odorant receptor 4-like	10
LOC107996764	odorant receptor 4-like	12
LOC107996743	odorant receptor 4-like	14
LOC107996763	odorant receptor 4-like	15
LOC107996720	odorant receptor 4-like	7
LOC108000540	odorant receptor 13a-like	5
LOC107994426	odorant receptor 13a-like	17
LOC107996760	odorant receptor 22c-like	10
LOC107999581	odorant receptor 63a-like	15
LOC107996752	odorant receptor 49b-like	17
LOC108002046	glutamate receptor ionotropic, kainate 2	17
LOC108000903	glutamate receptor ionotropic, NMDA 2C-like	3
LOC107997799	gustatory receptor for sugar taste 64f-like	10

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
