# Peer review of "Physiological Analysis and Transcriptome Analysis of Asian Honey Bee (Apis cerana cerana) in Response to Sublethal Neonicotinoid Imidacloprid"

_insects, 2020, doi:10.3390/insects11110753_

Round 1
Reviewer 1 Report
I read with attention the manuscript insects 958730 entitled
“ Physiological analysis and transcriptome analysis of Asian Honey Bee (Apis cerana cerana) in Response to Sublethal Neonicotinoids Imidacloprid” . It deals with the effects of neonicotinoids on Apis cerana. Even if these effect are widely studied on A. mellifera, they are poorly known on A. cerana. Authors rigorously studied different effects. The introduction gives a good overview of the current knowledge on this topic. the posed research questions can be better detailed. Moreover, even if the MS is well written there are many thing to reorder in the methods, as many information in the methods adopted are lacking. This makes the reader ask question about the tests conducted.
IN the following detail I ask some clarifications and give Authors some suggestions.
Line 65. Change olfactory with olfactory system
Line 72. Change letal level with letal dose
Line 88. Put A. mellifera in italicus A.mellifera
Line 119: please define better how the pollen was given and which pollen was used, the one in the frames, or fresh pollen?
Line 127: please provide 5 concentration of imidacloprid used
Line 138: change removed with moved
Line 150: please provide which concentration of imidacloprid was used. If you refer to the concentration previously decribed, evidence this aspect
Line 159-166: please write how many specimens have been used for each test. Beside the bees used from wich of the previous test arrived, which was the procedure of conservation? When were they collected. Like described for RNA isolation.
Results.
Line 254. I do not understand why fluvalinate was tested. Authors have never inserted fluvalinate in the methods, and now they give results about it. Please add information about fluvalinate tests and comparison with imidaloprid in the methods.
Line 255: authors write that fluvalinate is applied topically to bees, but then they compare with imidacloprid. In the methods imidacloprid is stated as ingested. Do they apply the same procedure to bee? And if so why? Please Justify in the methods
Title 3.1; Line 256; 258; 264 and many other times in the text and figures: please write cerana instead of cerena.
Line 257- 258: “The results indicated that imidacloprid is 257 more toxic than fluvalinate to A. cerana.” This consideration is not a result but should be moved in the discussion.
Line 258-259: “To mimic the acute sublethal effects of the imidacloprid from 258 contaminated beebread, we applied LC5 (0.968mg/L) of imidacloprid to 50% sucrose as the treatments 259 for the following experiments, which is closed to the maximum level (0.912 mg/L) in pollen obtained 260 from bee hives [39]” this are not result but methods, please reconsider them.
Line 262-266“ Previous studies have shown that pesticide contaminants can influence bees’ behavior, such as 262 climbing ability [40]. In this study, we firstly investigated the effects of imidacloprid on the geotactic 263 climbing ability of A. cerena. After received the sublethal dose of imidacloprid for 24 h, bees above 10 264 cm line at the end of the 15 second time period were counted and present as percentage of the number 265 of bees above the line as compared to the number of all the tested bees (Figure 1 A). “ again these are methods, not results. Results are limited to the following phrase.
Table 1: give evidence this is LD50. Change the title of the table.
Line 566-571 A. mellifera in Italicus
Author Response
Thank you for your useful comments and suggestions on the language and structure of our manuscript. We have modified the manuscript accordingly, and detailed corrections are listed below point by point:
Line 65. Change olfactory with olfactory system
We replaced this in our revised manuscript (line 66).
Line 72. Change letal level with letal dose
We changed this in our revised manuscript (line 73).
Line 88. Put A. mellifera in italicus A.mellifera
We are so sorry to make this mistake. It has been corrected in our revised manuscript.
Line 119: please define better how the pollen was given and which pollen was used, the one in the frames, or fresh pollen?
Thanks for the reviewer’s suggestion. We add the detailed information as replaced “pollen” by “pollen (collected from the apiary placed in the forest and no chemical pesticides were applied at the pollen collection time)” in line (124 -125) in the revised manuscript.
Line 127: please provide 5 concentration of imidacloprid used
Thanks for the reviewer’s suggestion. We have added the 5 concentration of imidacloprid in line 138, which are 1.8 mg/L, 2.5 mg/L, 3.5 mg/L, 5 mg/L, 7 mg/L, respectively.
Line 138: change removed with moved
It has been changed in our revised manuscript.
Line 150: please provide which concentration of imidacloprid was used. If you refer to the concentration previously decribed, evidence this aspect
The concentration used for sucrose responsiveness test was LC5. We have added the information in the revised manuscript.
Line 159-166: please write how many specimens have been used for each test. Beside the bees used from which of the previous test arrived, which was the procedure of conservation? When were they collected. Like described for RNA isolation.
Thanks for the reviewer’s suggestion. We have carefully re-written the methods section and added more detailed information in it. As the processing method of bees used in the enzyme activity assay has been described in “2.2 Pesticide exposure experiment” Line 146-149, we did not describe it in the later section. If you think it is necessary to do so, we may add a short description.
Results.
Line 254. I do not understand why fluvalinate was tested. Authors have never inserted fluvalinate in the methods, and now they give results about it. Please add information about fluvalinate tests and comparison with imidaloprid in the methods.
We want to propose that imidacloprid is much more toxic to A.cerana than fluvalinate. It is true that the fluvalinate bioassays are less related to the main topic of the present study. The authors agree to removed this result from the revised manuscript.
Thanks very much for the reviewer’s comments.
Line 255: authors write that fluvalinate is applied topically to bees, but then they compare with imidacloprid. In the methods imidacloprid is stated as ingested. Do they apply the same procedure to bee? And if so why? Please Justify in the methods
The fluvalinate related results had been delete from the the revised manuscript. Thanks again for the reviewer’s comments.
Title 3.1; Line 256; 258; 264 and many other times in the text and figures: please write cerana instead of cerena.
We are very sorry to give you such trouble for those spelling errors. They have been carefully corrected.
Line 257- 258: “The results indicated that imidacloprid is 257 more toxic than fluvalinate to A. cerana.” This consideration is not a result but should be moved in the discussion.
As the reviewer said, the fluvalinate bioassays is unnecessary in the present study. The authors decided to remove this result from the revised manuscript.
Line 258-259: “To mimic the acute sublethal effects of the imidacloprid from 258 contaminated beebread, we applied LC5 (0.968mg/L) of imidacloprid to 50% sucrose as the treatments 259 for the following experiments, which is closed to the maximum level (0.912 mg/L) in pollen obtained 260 from bee hives [39]” this are not result but methods, please reconsider them.
Thanks for the reviewer’s suggestion. We have carefully re-written the methods section and added these sentences in line 141-145.
Line 262-266“ Previous studies have shown that pesticide contaminants can influence bees’ behavior, such as 262 climbing ability [40]. In this study, we firstly investigated the effects of imidacloprid on the geotactic 263 climbing ability of A. cerena. After received the sublethal dose of imidacloprid for 24 h, bees above 10 264 cm line at the end of the 15 second time period were counted and present as percentage of the number 265 of bees above the line as compared to the number of all the tested bees (Figure 1 A). “ again these are methods, not results. Results are limited to the following phrase.
According to the reviewer indicated, we have removed these sentences.
Table 1: give evidence this is LD50. Change the title of the table.
Thanks for the reviewer’s suggestion. The Table 1 has been deleted from the revised manuscript the and the result of toxicity of imidacloprid removed to Method section.
Line 566-571 A. mellifera in Italicus
We have carefully fixed them.
Reviewer 2 Report
In my opinion, this is a high scientific report. The experimental plan is well drawn, the idea are clearly expressed and the conclusions are logical and derived from an efficient analyses of the results.
The main aim of the research is to expose Apis cerana to imidacloprid during different periods of time (1, 8, and 16 hours) and to subjected it to RNA sequencing 109, with the precise purpose to identify the transcriptomic changes. For my knowledge, there are a lot of studies regarding the effect of neonicotinoids on Apis genus, but only a little part of these studies have researched the sublethal effects of neonicotinoids on Apis cerana. The conclusions are good formulated, based on the results, which are valuable ones. The authors explained very well the results and the declared purpose of the study was fully achieved.
My only observation regards some titles from References sections: in some of them Latin names, like Apis melifera or Apis cerana are written in lower case and not in italics.
Overall, a very valuable work.
Author Response
In my opinion, this is a high scientific report. The experimental plan is well drawn, the idea are clearly expressed and the conclusions are logical and derived from an efficient analyses of the results.
The main aim of the research is to expose Apis cerana to imidacloprid during different periods of time (1, 8, and 16 hours) and to subjected it to RNA sequencing 109, with the precise purpose to identify the transcriptomic changes. For my knowledge, there are a lot of studies regarding the effect of neonicotinoids on Apis genus, but only a little part of these studies have researched the sublethal effects of neonicotinoids on Apis cerana. The conclusions are good formulated, based on the results, which are valuable ones. The authors explained very well the results and the declared purpose of the study was fully achieved.
My only observation regards some titles from References sections: in some of them Latin names, like Apis melifera or Apis cerana are written in lower case and not in italics.
Overall, a very valuable work.
Thanks very much for the reviewer’s comments. We have carefully corrected them in the text.
Reviewer 3 Report
This manuscript suffers from a lack of experimental details, a superficial understanding of toxicological principles, and a coherent take home message. The results and discussion are poorly organized, make tangential connections to the literature, and are highly speculative. There are too many DEGs that have questionable relationship to detoxification and insecticide sensitivity. Despite the statistical significance of these gene expression studies, the biological relevance of those differences is probably irrelevant.
Critical Comments
The first issue is that the talk about using sublethal concentrations of imidacloprid in this study. However, it is clearly stated the LC5 was used. At that concentration, you would expect to see 5% of the test subjects die. This expected mortality is not sublethal by definition as 5% of the population should die. it is stated on line 276 “no honey bee mortality was observed.” If this statement is true, then an LC5 was not used. An experimentally determined maximum sublethal concentration should have been used which is a concentration lower than the lower side of the 95% confidence interval of the LC1. This is done because an LC0 does not exist in toxicology.
Another aspect of this study that drew my attention were the bioassay results. The n column in table 1 is confusing. This should be the total number of bees used in the test across all doses. So if this is “40 bees were used per cage (L118)” and “five concentrations of imidacloprid (L127)”, then this is 720 insects including the presumed 120 tested in the control. If only 120 insects were used in the test, then I would have no faith in the parameters of the concentration response curves as those values are only present when a large number of animals are used in the bioassay. The highly precise numbers in the n column drew my attention because the exact numbers of animals in a bioassay are almost ever exactly what is intended and often you end up with odd numbers of actual animals in the test.
The slope of the concentration response curve for imidacloprid is suspiciously high. Literature values for the LC50 of imidacloprid in honey bees are 2.21 (Decourtye 2003 Pest Manag Sci 59:269), 1.1, 0.9, and 1.3 (Rinkevich 2015 Plos One 10: e0139841) 1.7, 1.66, 2.76, and 2.12 (Iwasa 2004 Crop Protection) to name a few. This trend generally holds for other species as well. This result seems extraordinary at first glance.
The fluvalinate bioassays are not described in the methods and only appear in the results. The are not mentioned in the discussion either. This is extremely unnecessary and tangential. Please remove this from the manuscript.
General Comments
Line 79 “Clothianidin acts as an additional stress factor by depressing the activation of NF-κB signaling 79 pathway…” but in figure 7, the expression of this gene is increased 5-fold at 16 hours post treatment. This discrepancy is not revisited.
L119 How much pollen was provided? What conditions were the emerged bees held at? The cage design and feeding protocols lack much needed details. Is there a reference for this protocol?
L133 How many bees were used from each of the 3 cages at each time point for enzyme analyses? It is unclear in the text. What storage conditions were the samples held at prior to analyses?
L175 What was used to generate the standard curve in this experiment to determine the nmol of product produced? A good amount of research shows that naphthyl acetate has very little to do with detoxification so its relevance here is questionable. On Line the 4-NPA abbreviation is out of place. The product is 1-naphtol and the substate typically noted as aNA or 1NA.
L194 Does this mean each sample was read in triplicate? Was it 3 cages used? Was it 3 bees used? The sample sizes are difficult to follow throughout this manuscript.
L267 Are you sure there was no mortality at 3 and 4 days in the imidacloprid treatment? Imidacloprid usually has higher toxicity the longer the exposure.
L277 Carboxylesterase activity is irrelevant to detoxification as imidacloprid does not have carboxylester bonds and synergist bioassays show no synergism of imidacloprid with esterase inhibitors. Furthermore, the time course of changes in detoxification are not consistent with the rate of detoxification as noted by changes in concentration of radiolabeled imidacloprid (Suchail 2004 Pest Manag Sci 60:1056).
L298 If detoxification rate of P450 and CarE are different after imidacloprid, where do the expression of those genes fall in the DEG classifications?
L317 This section is difficult to make sense of and seems extremely artificial. There does not seem to be a biological basis for these profiles. The relevance to any insight for detoxification is highly speculative at best.
L399 Why was the expression of CarE not evaluated especially when there was a difference in the activity at 4 hours? The P450s used in the validation test in Figure 7 are probably of very little relevance to detoxification. Manjon 2018 Current Biology 28:1 elegantly and unequivocally have demonstrated that CPY9Q1, 2, and 3 are exclusively responsible for imidacloprid metabolism in Apis mellifera and Bombus terrestris. Therefore, it is highly unlikely that orthologs of CYP6A14 or CYP304A1 is responsible for imidacloprid detoxification here.
L494 The expression of CYP6A14 and 304A1 decrease consistently over the time course using both methods. However, P450 activity increases at 4hrs in Fig 2. I would conclude these genes are irrelevant as per my previous comments.
L498 The topics brought up for discussion are not inclusive of all the results. For example, in figure 7 the expression of abecin increases to nearly 8-fold and defensin increases 12 fold after 16h. The conclusion I would make from that data is that imidacloprid induces immune gene expression. This is not discussed in the manuscript
L569-587 This passage is a great example of the speculative nature of the results and discussion in this study. Differences in gene expression need to be in physiological pathways or systems that are relevant to the behavior or phenotype in question. Without clear linkage of expression to phenotype, then results are speculative at best or Type I error is occurring.
L618 “The lower activity of P450 and CaE in exposed bees…” directly contradicts the results showing induction of both of those activities at 4h.
L621 “…carboxypeptidase Q-like genes…” are by definition not carboxylesterases. Therefore, this is irrelevant to CarE activity. Furthermore, later in this paragraph, CarE is not discussed.
Author Response
This manuscript suffers from a lack of experimental details, a superficial understanding of toxicological principles, and a coherent take home message. The results and discussion are poorly organized, make tangential connections to the literature, and are highly speculative. There are too many DEGs that have questionable relationship to detoxification and insecticide sensitivity. Despite the statistical significance of these gene expression studies, the biological relevance of those differences is probably irrelevant.
Critical Comments
The first issue is that the talk about using sublethal concentrations of imidacloprid in this study. However, it is clearly stated the LC5 was used. At that concentration, you would expect to see 5% of the test subjects die. This expected mortality is not sublethal by definition as 5% of the population should die. it is stated on line 276 “no honey bee mortality was observed.” If this statement is true, then an LC5 was not used. An experimentally determined maximum sublethal concentration should have been used which is a concentration lower than the lower side of the 95% confidence interval of the LC1. This is done because an LC0 does not exist in toxicology.
Another aspect of this study that drew my attention were the bioassay results. The n column in table 1 is confusing. This should be the total number of bees used in the test across all doses. So if this is “40 bees were used per cage (L118)” and “five concentrations of imidacloprid (L127)”, then this is 720 insects including the presumed 120 tested in the control. If only 120 insects were used in the test, then I would have no faith in the parameters of the concentration response curves as those values are only present when a large number of animals are used in the bioassay. The highly precise numbers in the n column drew my attention because the exact numbers of animals in a bioassay are almost ever exactly what is intended and often you end up with odd numbers of actual animals in the test.
The slope of the concentration response curve for imidacloprid is suspiciously high. Literature values for the LC50 of imidacloprid in honey bees are 2.21 (Decourtye 2003 Pest Manag Sci 59:269), 1.1, 0.9, and 1.3 (Rinkevich 2015 Plos One 10: e0139841) 1.7, 1.66, 2.76, and 2.12 (Iwasa 2004 Crop Protection) to name a few. This trend generally holds for other species as well. This result seems extraordinary at first glance.
The fluvalinate bioassays are not described in the methods and only appear in the results. The are not mentioned in the discussion either. This is extremely unnecessary and tangential. Please remove this from the manuscript.
Thanks very much for the reviewer’s comments.
We have carefully considered the reviewer’s comments about the sublethal dose that we used in this study. So far, there is no consistent definition of “sublethal” in toxicology studies of bees and other species. In many studies, researchers considered LC5 as an experimental concentration to study sublethal effects, (e.g. Xiao et al, 2016 Ecotoxicology; Ullah et al, 2020 Pestic Biochem Physiol.; Sobrino-Figueroa et al, 2014 J Environ Biol.). The LC5 of imidacloprid was theoretically calculated by a software(Polo Plus 2.0)and the concentration is 0.986 mg/L (0.4-1.449) with 95% confidence intervals in parentheses. We also calculated LC1 dose of imidacloprid which is 0.874 mg/L (0.226-1.312), with 95% confidence intervals in parentheses by the software. There is a clear overlap between LC5 and LC1 within 95% confidence intervals. Thus, even we used the lower side of the 95% confidence interval of the LC1, it also does not mean 1% bees die at this concentration. Since we want to observed more obvious sublethal effects in the physiological and molecular levels of bees, the selected concentration should be closer to the low lethal concentration rather than the non-lethal concentration.
We are sorry to the mis-expression and the confusing statements. 720 bees were used to conduct the survival experiment, including 120 bees in the control group. We have carefully re-written the methods section in line 133-136.
We also noticed that the LC50 dose of imidacloprid determined in this work is higher than many previous studies. The LC50 dose of imidacloprid determined in this work is above the considered “field realistic” range as well as the estimated LC50 dose of A. mellifera (1.76 mg/L), as we stated in the discussion. However, the pesticide resistance of bee populations in different regions and colonies varies greatly. Besides, different feeding methods and experiment conditions can also cause huge variation in the value of LC50 among different studies.
We want to propose that imidacloprid is much more toxic to A.cerana than fluvalinate. It is true that the fluvalinate bioassays are less related to the main topic of the present study. The authors agree to remove this result from the revised manuscript.
General Comments
Line 79 “Clothianidin acts as an additional stress factor by depressing the activation of NF-κB signaling 79 pathway…” but in figure 7, the expression of this gene is increased 5-fold at 16 hours post treatment. This discrepancy is not revisited.
Nuclear factor kappa B (NF-κB) plays an important role in the immune response in mammals and insects. It is normally found in an inactive form associated with regulatory proteins called inhibitors of κB(IkB). Cactus is an IkB that binds to NF-kB to repress its activity. In this study, we found the expression of two NF-kappa-B inhibitor that are cactus like (LOC107996104 and, LOC108002391, profile 16) were increased in exposed bees after 8 h imidacloprid exposure, as shown in figure 7 and Table 2 (the Table number has changed to Table 1 in the revised manuscript). To some extent, this result supports the hypothesis that neonicotinoids can depress the activation of NF-κB signaling pathway by enhancing the transcription of negative regulator of NF-κB in bees.
L119 How much pollen was provided? What conditions were the emerged bees held at? The cage design and feeding protocols lack much needed details. Is there a reference for this protocol?
Thanks for the reviewer’s comments. We have carefully re-written the methods section and added the reference. Honey bees were provided with pollen, water, and sucrose ad libitum for the first week and were then subjected to imidacloprid treatments. We used plastic feeders with eight 300 μl separate feed tubes to feed bees and the feeders were changed daily. The additional information of cages and storage condition were added in line 119-127 in the the revised manuscript.
L133 How many bees were used from each of the 3 cages at each time point for enzyme analyses? It is unclear in the text. What storage conditions were the samples held at prior to analyses?
Thanks for the carefulness of the reviewer. For the enzyme activity assay, abdomen or midgut collected form three bees were pooled together as one sample for each replicate (per cage for each replicate). We have re-written the methods section and add detailed information in line 175-182.
L175 What was used to generate the standard curve in this experiment to determine the nmol of product produced? A good amount of research shows that naphthyl acetate has very little to do with detoxification so its relevance here is questionable. On Line the 4-NPA abbreviation is out of place. The product is 1-naphtol and the substate typically noted as aNA or 1NA.
Thanks for the reviewer’s suggestion. The standard curve of α-naphthol was constructed by adding increasing amounts of α-naphthol (nmol) (range from 0.0005 to 0.05 mM) and measuring the increase in optical density at 600 nm as described in Chanda 1997, Fundamental and Applied Toxicology. The results were expressed in mmol of α-naphthol mg of protein−1 min−1. It should be a misspelling and we are sorry for the confusing statement. It has been corrected in the revised manuscript.
L194 Does this mean each sample was read in triplicate? Was it 3 cages used? Was it 3 bees used? The sample sizes are difficult to follow throughout this manuscript.
We have carefully re-written this part and made it more informative (line 175-182).
L267 Are you sure there was no mortality at 3 and 4 days in the imidacloprid treatment? Imidacloprid usually has higher toxicity the longer the exposure.
We are sorry to mis-express this. It has been corrected in the revised manuscript(line 282-284). It is logical that time and the duration of the exposure is a stressor and will cause higher toxicity. However, according to the experiment records, very few bees (1-2 bees per cage at maximum) died after the climbing test under the low dose of imidacloprid applied in this study. As we are focus on the short-term effect of imidacloprid on A. cerana, we only determined the toxicity of the imidacloprid for 24 hours. We did not measure the survival rate for 3 and 4 days. It is required another survival experiment to verify whether 5% of bees really die under this condition.
L277 Carboxylesterase activity is irrelevant to detoxification as imidacloprid does not have carboxylester bonds and synergist bioassays show no synergism of imidacloprid with esterase inhibitors. Furthermore, the time course of changes in detoxification are not consistent with the rate of detoxification as noted by changes in concentration of radiolabeled imidacloprid (Suchail 2004 Pest Manag Sci 60:1056).
CarEs are involved in the metabolic detoxification of dietary and environmental xenobiotics in insects. Previous studies have reported that sublethal dose of imidacloprid can influence the activity of carboxylesterase (CarE) in many insects, for example, honeybees (Li 2017, Pesticide Biochem and Physiol.) Sitobion avenae (Lu 2016 Bull Entomol Res.), Rhopalosiphum padi (Li 2018 PLoS One), and Aphidius gifuensis (Kang 2018 Front Physiol.). Moreover, recent transcriptomic investigation has revealed that expression of CarEs were also influenced by imidacloprid in A. mellifera (Ma 2018 Biochimie; Aufauvre 2014 PLoS One). In the present study, we carefully stated that sublethal dose of imidacloprid can affect the activity of this major detoxification enzyme. It is certainly an interesting topic about why and how CarEs respond to imidacloprid toxicity even though imidacloprid does not have carboxylester bonds in honey bees. However, this was not studied in the present work. Thank you for the sincere comments about this issue and further consideration should be taken in our future study.
Suchail et al.(2004) performed a nice work on pharmacodynamics of imidacloprid in A. mellifera. The imidacloprid half-life was 5 h after intoxication with a dose of 20 µgkg−1 bee. After total ingestion of 50 µg imidacloprid kg−1 bee, the Imidacloprid had an elimination half-life of 4.5h. It reflects that the concentration of imidacloprid exposed to bees is important for the elimination half-life of imidacloprid. They found that 5-hydroxyimidacloprid and olefin, two metabolites having toxicity close to that of imidacloprid are in low amounts compared with imidacloprid and present a peak value 4 h after oral exposure to 20 or 50µg kg−1 bee. According to our data, the activity of P450 and CarEs achieved the highest activity at 4 h (1.46 and 1.27-fold more than control, P<0.05), and then the enzyme activities were sharply decreased after exposure with the time duration (Figure 2 A & 2 B). Thus, this result is not completely conflicted with changes in concentration of radiolabeled imidacloprid. Besides, it is difficult to compared the detoxification rate of bees under different experimental conditions, especially from two different subspecies. As we state in the introduction, the effect of pesticides to A. mellifera cannot be completely inferred to other bee species.
L298 If detoxification rate of P450 and CarE are different after imidacloprid, where do the expression of those genes fall in the DEG classifications?
We identified five P450 isoforms which is differently expressed responding to imidacloprid exposure. five P450 genes belonging to CYP3 subfamily (LOC108000928, profile 1; LOC 107999234, profile 1) and CYP6 subfamily (LOC108004038, profile 1; LOC108003993, profile 7; LOC108003970, profile 9). Most of them are clustered into down-regulated pattern profile. Five genes encoding CarEs (LOC107993699, LOC108002181, LOC107993965, LOC107994233 and LOC107998239) were identified in the RNA-seq analysis. Two venom carboxylesterases (LOC107994233 and LOC107998239) showed down-regulated (-2.39 and-2.03 fold) after 16 h imidaclopride exposure but not classified as down-regulated in the trend analysis due to the FDR value greater than 0.064 and 0.073. We have added it in discussion in line 643-650.
L317 This section is difficult to make sense of and seems extremely artificial. There does not seem to be a biological basis for these profiles. The relevance to any insight for detoxification is highly speculative at best.
To investigate the major expression trends of DEGs over the three time points, we performed a cluster analysis using STEM software and found that all 709 DEGs were assigned to 17 clusters. This analysis integrates genes with similar expression trends based on the log2 ratio rather than relevant biological processes. Thus, genes with different functions may gather in an expression pattern. The similar analysis method was also reported in Chen 2018 plant science, Zhao 2018 Gene, etc.
L399 Why was the expression of CarE not evaluated especially when there was a difference in the activity at 4 hours? The P450s used in the validation test in Figure 7 are probably of very little relevance to detoxification. Manjon 2018 Current Biology 28:1 elegantly and unequivocally have demonstrated that CPY9Q1, 2, and 3 are exclusively responsible for imidacloprid metabolism in Apis mellifera and Bombus terrestris. Therefore, it is highly unlikely that orthologs of CYP6A14 or CYP304A1 is responsible for imidacloprid detoxification here.
Although the expression of two venom carboxylesterases were decreased after 16 h imidacloprid exposure (2.39 and-2.03 fold), statistical analysis show that the difference is not significant due to the FDR value greater than 0.064 and 0.073. Thus, we didn’t perform qRT-PCR with these genes in the validation test.
We agree with the reviewer that the three members CPY9Q1, 2, and 3 in CPY3 clade is very essential for metabolize imidacloprid in A.mellifera. Whether the differential expressed P450s identified in our RNA-seq analysis have detoxification functions or simply involved in other metabolic pathways was not verified in this study. It has been suggested that the expression of CYPs is a complex phenomenon involving transcriptional factors, and the overlap of regulatory pathways. Hence, we can’t draw a conclusion that the P450s identified here have absolutely nothing to do with the imidacloprid resistance in A.cerana. It will, therefore, be particularly interesting to further characterize transcriptional elements in A.cerana P450 genes and regulatory factors involved in the insecticide resistance in order to fully understand the molecular mechanisms involved. We will take this issue as our future research aim. Thanks for the sincere comments. We have added it in discussion in line 654-658.
L494 The expression of CYP6A14 and 304A1 decrease consistently over the time course using both methods. However, P450 activity increases at 4hrs in Fig 2. I would conclude these genes are irrelevant as per my previous comments.
Thanks for the comments. We are not sure how much the expression changes of the two P450s genes affect the entire activity of the enzyme. Besides, the correlation between mRNA and protein levels keeps unknown. Considering the complex physiological response of honey bees to abiotic stressors,in addition to the level of expression, some other key factors such as the slope of response, genes specificity and redundancy, and post-translational modification should be considered to accurately assess the gene expression. It may be not very rigorous to draw a conclusion that the P450s identified here have absolutely nothing to do with the imidacloprid resistance in A.cerana.
L498 The topics brought up for discussion are not inclusive of all the results. For example, in figure 7 the expression of abecin increases to nearly 8-fold and defensin increases 12 fold after 16h. The conclusion I would make from that data is that imidacloprid induces immune gene expression. This is not discussed in the manuscript
The genes shown in Figure 7 were randomly selected to verify the accuracy of RNA-seq data, thus, not all individual genes were discussed in the manuscript.
L569-587 This passage is a great example of the speculative nature of the results and discussion in this study. Differences in gene expression need to be in physiological pathways or systems that are relevant to the behavior or phenotype in question. Without clear linkage of expression to phenotype, then results are speculative at best or Type I error is occurring.
Thanks for the reviewer’s comments. We have to admit the current limitations of omics research, especially in transcriptomic and proteomic studies. It becomes apparent that the correlation between mRNA and protein levels was remarkably low. The low correlation is generally hypothesized to result from post-translational modification or the difficulties associated with the annotation of differentially expressed proteins relevant correspondences. This makes it difficult to establish a direct correlation between the differences in gene expression in transcriptomics and physiological functions. This issue exists not only in this study, but also in others. But we still want to figure out some valuable information form the large-scale data. For instance, we found AMPs and most genes in Toll pathway were classed in profile 12, which suggested that the cellular immune-related pathway was not Immediately activated after imidacloprid exposure.
L618 “The lower activity of P450 and CaE in exposed bees…” directly contradicts the results showing induction of both of those activities at 4h.
We have corrected the sentence with “The activity of CaE achieved were also depressed after 4 hours of exposure to imidacloprid” in line 643.
L621 “…carboxypeptidase Q-like genes…” are by definition not carboxylesterases. Therefore, this is irrelevant to CarE activity. Furthermore, later in this paragraph, CarE is not discussed.
Considering the reviewer’ s suggestion, we have rewritten this section and add a short discussion about CarE in line 636-650.
Round 2
Reviewer 1 Report
The Authors have slighlty improved the manuscript including all the preciphic requested. I appreciated their work and their results. I think it can be accepted.
Best regards